

# Technical note: A low cost, automatic soil-plant-atmosphere enclosure system to investigate CO₂ and ET flux dynamics.

Wael Al Hamwi[1], Maren Dubbert[1], Jörg Schaller[2,3], Matthias Lück[1], Marten Schmidt[1], Mathias Hoffmann[1]

[1]Isotopes Biogeochemistry and gas fluxes, Leibniz-Center for Agricultural Landscape Research (ZALF), Müncheberg, 15374, Germany.

[2]Silicon Biogeochemistry, Leibniz-Center for Agricultural Landscape Research (ZALF), Müncheberg, 15374, Germany.

[3]Department of Agricultural Sciences, Nutritional Sciences, and Environmental Management, University of
Giessen, 35390, Giessen, Germany.

*Correspondence to:* Wael Al Hamwi (Wael.Alhamwi@zalf.de)

## Abstract

Investigating greenhouse gases (GHG) and water flux dynamics within the soil-plant-atmosphere-interphase is a
key for understanding ecosystem functioning, as these dynamics reflect the ecosystem´s responses to environmental changes. Understanding these responses is hence essential for developing sustainable agriculture systems that can help to adapt to global challenges such as inter-alia increased drought. Typically, an initial understanding of GHG and water flux dynamics is gained through laboratory or greenhouse pot experiments, where gas exchange is often measured using commercially available, manual closed (leaf) chamber systems.
However, these systems are usually rather expensive and often labor-intensive, thus limiting the number of different treatments that can be studied and their repetitions. Here, we present a fully automatic, low cost (<1.000 Euro), multi-chamber system based on Arduino, termed "greenhouse coffins". It is designed to continuously measure canopy CO₂ and evapotranspiration (ET) fluxes. And it can operate in two modes: an independent and a dependent measurement mode. The independent measurement mode utilizes low cost NDIR CO₂ (K30 FR) and
relative humidity (SHT31) sensors, thus making each "greenhouse coffin" a fully independent measurement device. The dependent measurement mode connects multiple "greenhouse coffins" via a low cost multiplexer (< 250 Euro) to a single infrared gas analyzer (LI-850, LI-COR Inc., Lincoln, USA), allowing for measurements in series, achieving cost efficiency, while also gaining more flexibility in terms of target GHG fluxes (potential extension to N₂O, CH₄, stable isotopes). In both modes, CO₂ and ET fluxes are determined through the respective
concentration increase during closure time. We tested both modes and demonstrated that the presented system is able to deliver precise and accurate CO₂ and ET flux measurements using low cost sensors, with an emphasis on calibrating the sensors to improve measurement precision. Through connecting multiple greenhouse coffins via our low cost Multiplexer to a single infrared gas analyzer in the dependent mode, we could show moreover that the system can efficiently measure CO₂ and ET fluxes in a high temporal resolution across various treatments with
both labor and cost efficiency. Therefore, the developed system offers a valuable tool for conducting greenhouse experiments, enabling comprehensive testing of plant-soil dynamic responses to various treatments and conditions.



## 1. Introduction

Climate change, leading to rising temperatures, altered precipitation patterns, and more frequent extreme weather events, is threatening ecosystem functioning globally (Ummenhofer and Meehl 2017). Agricultural systems are particularly vulnerable towards extreme weather events such as droughts and heat waves (Altieri et al. 2015). To cope with these challenges, managing agricultural systems to sustainably use their water resources while maximizing carbon sequestration is critical. Moreover, agriculture significantly contributes to greenhouse gas (GHG) emissions (Tubiello et al. 2013; Chataut et al. 2023). At the same time, a transfer to sustainable agricultural practices offers substantial potential for Climate Change mitigation through, e.g., decreased GHG emissions or increased soil carbon sequestration (Lal 2004; Powlson et al. 2016). Understanding soil-plant-atmosphere feedbacks of $CO_2$ (and other GHG emissions) as well as water fluxes is therefore crucial to understand the functioning of soil-plant systems under environmental stress as well as to develop effective mitigation strategies (Zhang et al. 2002; Joshua B. Fisher et al. 2017).

Chamber-based systems (automatic or manual) in conjunction with high temporal resolution gas analyzers are one of the most common techniques for directly measuring $CO_2$ and evapotranspiration (ET), providing precise data on a leaf to plot scale (Smith et al. 2010; Dubbert et al. 2014; Riederer et al. 2014). However, especially their labor intensive usage (manual chambers) and high costs associated with commercially available gas analyzers, multiplexers, and Automatic or semi-automatic chamber systems are significant barriers to extensive research, particularly in developing countries (Martin et al. 2017; Blackstock et al. 2019; Savage and Davidson 2003). This limits measurements to specific locations (Savage and Davidson 2003), predominately in the global north, thus preventing a holistic collection of data related to environmental issues worldwide, thereby impeding complete understanding and effective address at a global level (Macagga et al. 2024). Especially experiments at the mesocosm scale enable (semi-) controlled experimental setups in a greenhouse or climate chamber, with the opportunity to manipulate environmental conditions, thus enabling the exploration of the impact of isolated environmental treats. In doing so, greenhouse experiments bridge the gap between single plant laboratory and field studies, facilitating a more nuanced understanding of ecological dynamics (Riebesell et al. 2013; Stewart et al. 2013).

In recent years, low cost devices for environmental research with a do-it-yourself (DIY) approach have been increasingly developed to overcome the cost barrier associated with high-cost commercial chamber-based gas-exchange systems (Fisher and Gould 2012; D'Ausilio 2012). These DIY systems leverage affordable microcontrollers and sensors to build custom measurement tools designed for specific research needs. By integrating sensors for $CO_2$ and/or ET with microcontrollers, researchers were able to develop portable, precise, and cost-effective devices for monitoring $CO_2$ and ET fluxes, such as Macagga et al. (2024) and Bonilla-Cordova et al. (2024). Others went a step further and developed fully automated measurement systems to determine $CO_2$ efflux, such as the "Fluxbots" (Forbes et al. 2023). However, most "low cost" chamber developments target towards the in-situ use, while low cost developments for soil-plant enclosures at the mesocosm scale are scarce.

To fill this gap, we developed and validated the "Greenhouse Coffins", a novel low cost automatic soil-plant enclosure system, designed to monitor $CO_2$ and ET fluxes within greenhouse experiments in a fully automatic manner. We hypothesize that 1) a single "greenhouse coffin" employing low cost sensors can measure $CO_2$ and ET fluxes accurately and reliably, comparable to a high-cost gas analyzer. 2) By combining several "greenhouse



coffins" and adding a low cost self-constructed multiplexer, we are able to monitor gas fluxes via one infrared gas analyzer for different treatments cost-efficiently. To test these hypotheses, we performed a number of experiments validating the different components of the greenhouse coffins. Additionally, we evaluated the accuracy and

precision of used low cost NDIR $CO_2$ (K30 FR) and RH sensors (SHT31) by comparing their based calculated $CO_2$ and ET fluxes with results obtained with a commercial infrared gas analyzer (LI-850, LI-COR Inc., Lincoln, USA). Furthermore, we tested the system's ability to link multiple greenhouse coffins to one gas analyzer using a low cost multiplexer.

## 2. Material and methods

### 2.1. Hard and software implementation

The system "Greenhouse Coffins" consists of one to multiple enclosed transparent chambers (PVC; 180*40*60 cm) that can house an entire soil-plant-atmosphere system (Fig. 1). Each chamber can be accessed through a front door sealed using a rubber rope. The front door is equipped with a sliding window mechanism, which is opened and closed by a linear actuator. The sliding window covers two openings, behind which two opposing directed 9V

axial fans are installed to allow for quick air exchange while the window is open. Ventilation within each chamber is enabled through two additional axial fans at the bottom and top of the door. Each chamber is operated individually by a control unit consisting of an Arduino Uno-like microcontroller (ATmega 328-Board) with an attached logger shield module. This module is equipped with an SD card reader and 2 GB SD card for data storage, along with a real-time clock (RTC) ensuring accurate timekeeping while off power. For easy data access, a

Bluetooth module is connected to the microcontroller. To steer the opening and closure of the sliding window, the microcontroller switches a double relay, which is connected to the linear actuator (Fig. 2). During the closure of the sliding door, the two axial fans behind it are switched off via a Mosfet (IRLZ44N) connected to two resistors (200 and 10000 Ω). The power supply for each Greenhouse Coffin system is provided by a 9 V charger, connected to the microcontroller and axial fans, as well as a linear actuator (requiring 12 V) through a DC-DC buck boost

power converter. The control unit is fitted in an outdoor waterproof sealed box (19*12*5 cm) in the top-right corner of the door. When operated independently (independent mode), each greenhouse coffin utilizes a low cost NDIR-based $CO_2$ (0-10000 ppm, ±30 ppm ±3 % accuracy; K30 FR, Senseair AB, Sweden) and an air humidity and temperature sensor (SHT31, ±2% accuracy, Sensirion AG, Switzerland) placed on the inner side of the door (Macagga et al. 2024).

The individual coffins can be operated together by connecting multiple greenhouse coffins with a low cost multiplexer unit. This multiplexer unit switches a series of normally closed solenoid valves acting as air inlets and outlets, thus enabling to chain each greenhouse coffin with a single gas analyzer. The Multiplexer is controlled by an Arduino mega-like microcontroller (ATmega 2560), which controls a 16-fold relay model. Each of the 16 relays is linked to two solenoid valves, which open and close the air inlet and outlet of a greenhouse coffin. Relays are

operated in series. When a relay is powered up, the normally closed solenoid valves connected to it open, connecting the greenhouse coffin to the gas analyzer in a closed loop. A voltage sensor connecting the two solenoid valves and the control unit of the greenhouse coffin, signalizes when the solenoid valves are open and the sliding door needs to be closed to conduct measurements, thus enabling indirect communication and synchronization between the Multiplexer and each attached greenhouse coffin. The greenhouse coffin measured is indicated by an

LCD display connected to the Arduino mega-like microcontroller of the multiplexer unit. A Bluetooth module





(HC-05) allows for easy data access. The power supply for the 16-fold relay is provided by a 12 V charger, connected to a Boost converter step up/down (HW-140 DC-DC), adjusting the energy to 9 V for the microcontroller. Figure 3 shows the assembled connection of the different components. To enable remote access to the system during 24/7 measurements, a second Bluetooth module connected to an Arduino Uno-like

microcontroller with a logger shield acts as an uplink station. When connected to a stationary PC with internet access, incoming data transferred between both Bluetooth modules can be accessed on time using remote access software (e.g., Anydesk). Detailed information on component prices and distributors for both modes can be found in Table 1. The software was developed using Arduino IDE 2.0.0.





**Figure 1: sketch illustrates the greenhouse coffins system. (left) The independent mode, a single unit comprises (1) the chamber body, (2) the front door, (3) the control unit, (4) the linear actuator, (5) the sliding door, (6) ventilation fans, and (7) air mixing fans. (right) the dependent mode consists of (1) multiple greenhouse coffins and (2) a low cost multiplexer connected to a single gas analyzer.**




**Table 1 Components needed to construct one "Greenhouse Coffin" and a multiplexer, respectively. The prices are based on orders placed on Juli 30, 2023.**

| | Component | Amount | Price € | Distributor |
|---|---|---|---|---|
| **Greenhouse Coffin** | Chamber from pvc material (180*40*60 cm) | 1 | 600 | www.romid.pl |
| | Arduino uno like microcontroller (ATmega 328-Board) | 1 | 3.11 | www.az-delivery.de |
| | Datalogger module | 1 | 1.14 | |
| | Boost converters step up/down (HW-140 DC-DC) | 1 | 5 | |
| | 2-Relay module | 1 | 3 | |
| | Bluetooth module (HC-05) | 1 | 4.99 | |
| | Outdoor box (170*110*48 mm) | 1 | 13.98 | www.amazone.de |
| | Pvc hard foam plate | 1 | 1 | |
| | 0.5 mm2/20 awg electrical wire,7 colors | 1 | 2.5 | |
| | Luster terminals | 8 | 0.07 | |
| | Mosfet ( IRLZ44N) | 1 | 0.79 | |
| | SD MEMORY CARD (2 GB.10 MB/s) | 1 | 6 | |
| | 8 pin aviation connectors | 1 | 1.46 | |
| | Power jack socket | 2 | 1.49 | |
| | 8 Core cable (1 m) | 1 | 3.5 | |
| | Rubber rope (1.5 m) | 1 | 0.73 | |
| | Self-adhesive hooks | 20 | 0.41 | |
| | $CO_2$ sensor (Senseair k30 FR) | 1 | 85 | www.senseair.com |
| | RH and temperature sensor (SHT31) | 1 | 6.42 | www.aliexpress.com |
| | Electric linear actuator | 1 | 19.50 | |
| | Power supply 9v adapter | 1 | 9.10 | www.reichelt.de |
| | Axial fan | 4 | 3 | |
| | **Sum** | | **791.96** | |
| **Multiplexer** | 16-channel relay module 12 V | 1 | 10 | www.az-delivery.de |
| | Boost converters step up/down ( HW-140 DC-DC) | 1 | 5 | |
| | Bluetooth module (HC-05) | 1 | 4.99 | |
| | Arduino Uno-like microcontroller (ATmega 328-Board) | 1 | 3.11 | |
| | Arduino mega-like microcontroller (ATmega 2560) | 1 | 9.09 | |
| | LCD display with I2C interface | 1 | 5.49 | |
| | Datalogger module | 1 | 1.14 | |
| | B&W outdoor case typ1000 | 1 | 39.74 | www.amazone.de |
| | Voltage detection sensor | 1 | 1 | |
| | Power switch | 2 | 0.8 | |
| | Power jack socket | 13 | 1.49 | |
| | 2 ports 1/4 normally closed pneumatic control valve | 12 | 9.49 | |
| | 0.5 mm2/20 awg electrical wire,7 colors | 1 | 2.5 | |
| | **Sum** | | **216.91** | |






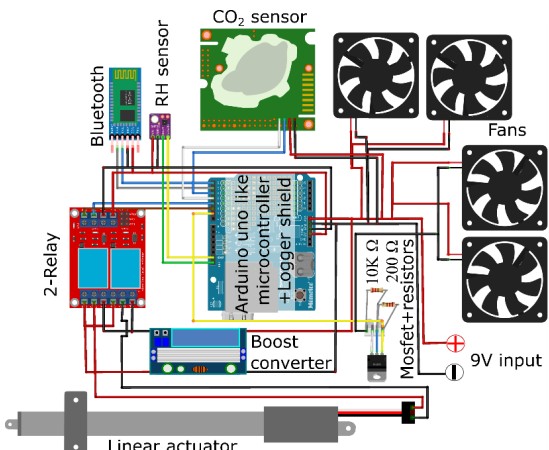

**Figure 2 Schematic representation of the wiring of one Greenhouse Coffin in the dependent mode.**

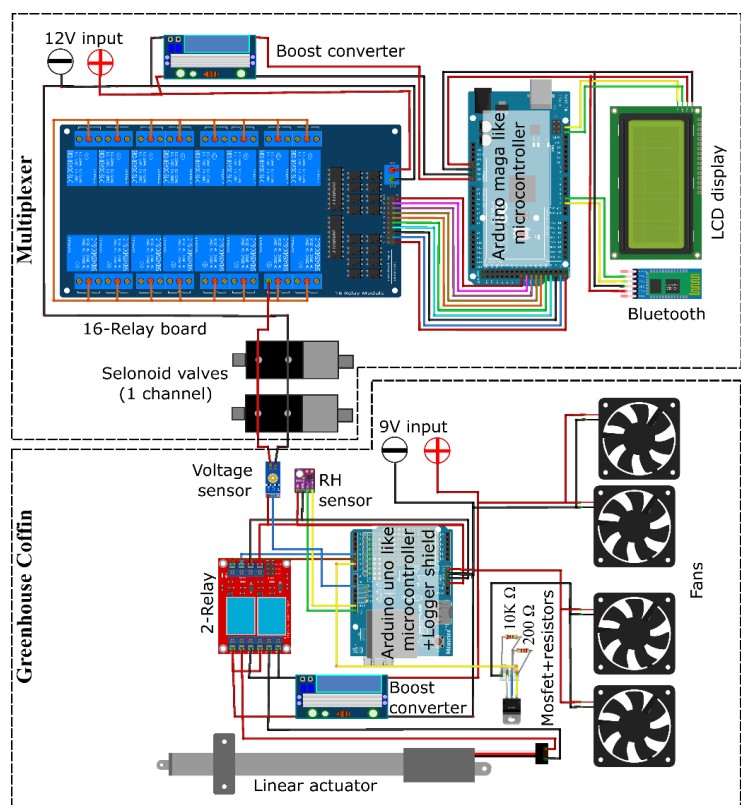

**Figure 3 Schematic representation of the wiring of the Multiplexer (on the top) connected to one Greenhouse Coffin (on the bottom) in the independent mode.**




### 2.2. Sealing test

In total, three sealing tests were performed to check for any leakage from different components of the presented system. Sealing tests included evaluation of the airtightness of 1.) the sealing design of the door (without sliding window), 2.) the suitability of the sliding window to sufficiently seal the greenhouse coffin when closed and exchange air when open, 3.) the proper sealing of the solenoid valves of the Multiplexer.

To assess the airtightness of the door sealing design of the Greenhouse Coffin, a smoke bomb was used as suggested by (Hoffmann et al. 2018) and which was also used by (Olfs et al. 2018) for the leakage test on their developed chamber design to measure nitrous oxide. The smoke bomb was placed inside the Greenhouse Coffin and lit, which was then sealed for a 15 minute observation period. Any smoke escaping during this time indicated possible leaks.

To check for suitability of the sliding window to sufficiently seal the greenhouse coffin when closed and exchange air when open in its final setup (complete hardware implementation), distinct amounts of technical gas containing 1,000,000 ppm $CO_2$ ranging from 15 to 50 ml were repeatedly injected into its sealed headspace using a syringe. By connecting an infrared $CO_2$ gas analyzer (LI-850, LI-COR Inc., Lincoln, USA) to the inlet and outlet of the sealed Greenhouse Coffin, the $CO_2$ concentration change from before to after injection ($\Delta CO_2$ in ppm) was subsequently measured. To assess the absence of cross-contamination within the dependent mode, out of the six greenhouse coffins connected to the $CO_2$ gas analyzer (LI-820, LI-COR Inc., Lincoln, USA) through the solenoid valves for the inlets and outlets, five were each equipped with plants changing the headspace $CO_2$ concentration during measurements while one remained empty. In the following, this one system was then measured and checked for potential $CO_2$ concentration changes. The measurements were performed for each of the six connected Greenhouse Coffins.

### 2.3. Validation experiment

For the independent mode, to Test the accuracy and precision of the low cost sensors (K30 FR and SHT31) as well as the capability of the greenhouse coffins system, a greenhouse experiment was conducted. Therefore, two pots planted with sorghum were placed inside a greenhouse coffin. For non-stop 5 days with a 30-minute chamber closure frequency and 5 min chamber closure time, we measured the $CO_2$ and ET concentration using both low cost sensors (K30 FR and SHT31) and an infrared gas analyzer (LI-850, LI-COR Inc., Lincoln, USA), resulting in ~48 $CO_2$ and ~48 ET fluxes per day.

For the dependent mode, to test the ability of the system to continuously monitor $CO_2$ and ET fluxes across various treatments in a fully automated manner using a single gas analyzer, we connected six greenhouse coffins (two empty, two with Sorghum plants, and two with Maize plants) to a single infrared gas analyzer (LI-850, LI-COR Inc., Lincoln, USA) via a low cost multiplexer. Subsequently, the $CO_2$ and ET concentrations were measured for each of the six chambers. Similarly to the independent mode, measurements were conducted for non-stop 5 days with a 30 minute chamber closure frequency and 5 minute chamber closure time, resulting in ~48 $CO_2$ and ~48 ET fluxes per day and Greenhouse Coffin. The environmental variables inside the greenhouse (air temperature, relative humidity, and photosynthetically active radiation (PAR)) were obtained from the greenhouse's climate station.





### 2.4. Data processing

#### 2.4.1. CO₂ and ET calculations

The first and last 10% of each $CO_2$ and ET measurement were removed to exclude any potential noises from turbulence and pressure fluctuations during the closing and opening of the sliding window (Hoffmann et al. 2015). Additionally, the $CO_2$ concentrations measured with the LI-850 were corrected to the changes in water vapor during chamber measurements (Webb et al. 1980; McDermitt et al. 1993). The Relative humidity (RH) provided by the low cost sensor (SHT31) required to be converted to mass concentration following Hamel et al. (2015) Eq. (1) :

$$H_2O = \frac{RH.e^s}{100.P} ,\qquad(1)$$

where RH is relative humidity, es is saturated vapor pressure calculated according to (Allen et al. 1998), and P is gas pressure (Pa). Modular R scripts, as described by (Hoffmann et al. 2015) for $CO_2$ and (Dahlmann et al. 2023) for ET, were used to calculate $CO_2$ and ET fluxes measured during the validation experiment. $CO_2$ and ET fluxes were calculated using the ideal gas law and using a linear regression approach Eq. (2):

$$f = \frac{M.p.V}{R.T.A} \cdot \frac{\Delta c}{\Delta t} ,\qquad(2)$$

where M is the molar mass of the gas (g mol⁻¹), p is the ambient air pressure (Pa), V is the chamber volume (m³), R is the gas constant (8.314 m³ Pa K⁻¹ mol⁻¹), T is the temperature inside the chamber (K), A is the basal area (m²), and $\Delta c/\Delta t$ represents the linear concentration changes in $CO_2$ (e.g., Leiber-Sauheitl et al., 2014) and $H_2O$ over time (e.g., Dahlmann et al., 2023). A variable moving-window (window size 0.5 to 5 min) was applied to each chamber measurement to obtain the variables T and $\Delta c/\Delta t$. Accordingly, resulting multiple ET and $CO_2$ fluxes per measurement (using the generated variable moving window data subset) were evaluated based on specific criteria, including fulfilled prerequisites for applying a linear regression (normality (Lilliefors adaption of the Kolmogorov–Smirnov test), homoscedasticity (Breusch–Pagan test) and linearity), (2) regression slope ($p \le 0.1$, $t$ test), (3) range of within-chamber air temperature not larger than ±1.5 K and a PAR deviation (only for day measurements) not larger than ±20 % of the average to ensure stable environmental conditions within the chamber throughout the respective measurement window, and (4) no outliers present (±6 times the interquartile range(IRQ)). Calculated $CO_2$ and ET fluxes meeting all criteria were retained. In cases where multiple fluxes per measurement met all criteria, the $CO_2$ and ET fluxes with the steepest slope and closest timing to chamber closure were selected.

#### 2.4.2. Statistical analysis

The statistical analysis was done using Scipy and Sklearn packages in Python (version 3.9.12). To determine the suitable statistic test for the collected data during the laboratory validation and greenhouse trial, a Kolmogorov-Smirnov test ($p<0.05$) to assess the normal distribution was carried out. A pairwise Wilcoxon signed-rank was employed to determine the significance of the $CO_2$ and ET fluxes measured by the low cost NDIR sensor and LI-850 sensor, as well as to determine the significance of the $CO_2$ concentration measured during the cross-contamination test. A concordance correlation coefficient was employed to determine the accuracy of the low cost sensors, while the precision was determined by Root mean square error (RMSE) and Pearson correlation. The error calculation for $CO_2$ fluxes was quantified using a comprehensive error prediction algorithm described in detail by (Hoffmann et al. 2015) using R software (version 3.6.1). The approaches utilize bootstrapping alongside





$k$-fold subsampling to estimate uncertainties for each flux measurement and subsequent $R_{eco}$ and GPP parameterization. This approach was adapted to calculate the error for ET fluxes by (Dahlmann et al. 2023).

### 3. Results and Discussion

### 3.1. Sealing test

During our smoke bomb test, no visible leakage was detected. This test indicates the airtightness of the sealing between the coffin and its door, as well as the sliding window within the coffin´s door. However, a properly sealed coffin does not mean that its ventilation system is also sufficient. For repeated measurements, it is essential to

replace the chamber headspace air after each measurement, thus recreating atmospheric starting $CO_2$ and $H_2O$ concentrations. To test for this, we compared the $CO_2$ starting concentrations (n=26) obtained during the gas injection test. With an average of $420 \pm 1.82$ ppm, the $CO_2$ starting concentrations were not only close to the atmospheric $CO_2$ concentration (419.3 ppm, NOAA, 2023) but also showed a minor variation, with a minimum and maximum $CO_2$ starting concentration of 417 ppm and 425 ppm, respectively. Additionally, no significant

difference (pairwise Wilcoxon signed-rank, $p > 0.01$) between $\Delta CO_2$ measured by the LI-850 and the calculated mixing ratio was observed (Fig.4). Hence, the gas injection test evidenced the coffin ventilation system's effectiveness and overall airtightness when used in independent mode. While the coffins are properly sealed, insufficient solenoid valve closure could lead to cross-contamination, with concentration increases in one coffin affecting another. To test for proper sealing of the coffins and the connected Multiplexer when used in dependent

mode, the cross-contamination test was hence performed. In case of no cross-contamination, an empty coffin should show no concentration change during its measurement, irrespective of plants being present in the other coffins connected to the Multiplexer. When comparing the measured $\Delta CO_2$ concentration of the performed cross-contamination, no significant difference to 0 was found (pairwise Wilcoxon signed-rank test, $p > 0.05$). These results show that no cross-contamination due to the Multiplexer occurred within the independent mode. However,

since solenoid valves have moving parts that can show wear and tear with long-term use, it is recommended to repeat the cross-contamination test periodically and change the non-tight valve.

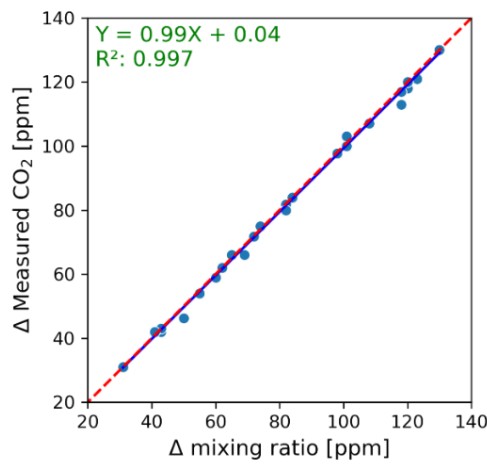





**Figure 4:1:1 agreement between the mixing ratio and the measured $\Delta$ CO$_2$ concentration change expressed as in ppm, was obtained during the laboratory validation.**

### 3.2. Validation experiment

### 3.2.1. Independent mode

The performed validation experiment, testing a single greenhouse coffin in independent mode, proved that by employing low cost sensors, $CO_2$ and ET fluxes can be fully automatized measured in a reliable and accurate manner. During the non-stop, 5 day validation experiment for the independent mode, the tested single greenhouse coffin and used low cost sensors showed no problems and worked reliably. Thus, out of 223 conducted automatic

measurements, more than 99% passed the flux calculation algorithm for $CO_2$ and ET, respectively. A rate only slightly below the 100% of $CO_2$ and ET fluxes passing flux calculation when using the LI-850 for $CO_2$ and $H_2O$ concentration measurements. Low cost as well as LI-850 derived $CO_2$ and ET fluxes showed mainly identical diurnal pattern, with low fluxes during nighttime and higher ET fluxes and a strong $CO_2$ uptake during daytime (Fig.5). Observed diurnal pattern, clearly followed monitored environmental parameters with higher ET fluxes

and $CO_2$ uptake with higher PAR and a higher $R_{eco}$ (nighttime measurements) with higher air temperatures. Figure 6 shows the 1:1 agreement and correlation between a) calculated $CO_2$ and b) ET fluxes based on low cost and LI-850 measurements of $CO_2$ and $H_2O$ concentrations as well as RH. The overall accuracy of both low cost sensors derived $CO_2$ and ET fluxes, is indicated by the high concordance correlation coefficient of 0.98 and 0.98 for $CO_2$ and ET, respectively. These results align well with the findings of (Macagga et al. 2024), who tested the accuracy

and suitability of the same sensors for in-situ manual closed chamber measurements and suggested a high degree of precision (scatter of measurement) and trueness (proximity to the true value) of used low cost sensors. Also, other studies highlighted the precision and trueness of the K30 FR and SHT31 sensors (Ali et al. 2016; Martin et al. 2017; Cannon et al. 2022). This is so far important, as a low trueness level can result in significant deviations from the actual value, while low precision can introduce noise/scatter into flux measurements (Werle 2011) and

might hamper repeatability of measurement results. However, while a RMSE of 5.05 µmol m$^{-2}$ s$^{-1}$ and 2.84 mm d$^{-1}$, and Pearson correlation coefficient of 0.99 and 0.98 proved the high precision for $CO_2$ and ET measured within this study, respectively, this was not equally the case for the trueness. On the one hand, calculated fluxes derived from $CO_2$ concentration and RH measurements using the low cost sensors correlated nearly perfectly ($R^2$: 0.98) with $CO_2$ and ET fluxes calculated based on LI-850 measurements of $CO_2$ and $H_2O$ concentrations (Fig. 6).

On the other hand, a clear underestimation in the case of higher $CO_2$ uptake by plants and ET flux rates for the low cost sensors is evident in Fig. 6. This is confirmed by conducted pairwise Wilcoxon signed-rank tests, which resulted in significant differences between fluxes calculated based on measurements with the K30 FR, SHT31 and LI-850 sensor, respectively ($p < 0.05$).

This underestimation, while potentially relevant for calculating fluxes, was not detected in other studies such as

inter-alia (Macagga et al. 2024), which is likely due to the considerably larger observation range in this study, when compared to previous studies (Ali et al. 2016; Cannon et al. 2022; Macagga et al. 2024). For example, Macagga et al. (2024) report a flux range of -17.05 to 13.74 µmol m$^{-2}$ s$^{-1}$ For $CO_2$ and 1.2 to 3.0 mm d$^{-1}$ for $H_2O$, while in our study, the $CO_2$ and ET flux amplitude was up to six times higher with $CO_2$ and ET fluxes ranging from -89.06 to 15.37 µmol m$^{-2}$ s$^{-1}$ and 0.7 to 18.66 mm d$^{-1}$, respectively. Since the underestimation was much

more pronounced at higher $CO_2$ uptake and ET, the trueness for the flux range of 20 to -30 µmol m$^{-2}$ s$^{-1}$ for $CO_2$ and range of 0 to 5 mm d$^{-1}$ was hence, comparable with findings of Macagga et al. (2024).



This highlights the importance to assess sensor performance for a wide range of concentrations during validation experiments when aiming to use low cost sensors and the independent mode to obtain accurate results. Especially, in case of $CO_2$ these experiments should not only consider conditions above but also below ambient. However,

the precision of the used low cost sensors throughout the entire flux measurement range for $CO_2$ as well as ET enabled us to derive and apply a correction function. After applying the correction function ($CO_2$: Y=1.11X-1.46, ET: Y=1.08X+0.04) on low cost sensor-based $CO_2$ and ET fluxes, cumulated $CO_2$ and ET fluxes for the 5 day validation experiment period (e.g. 5 day flux balance), derived using low cost and LI-850 measurements, differed by < 1.5 %.

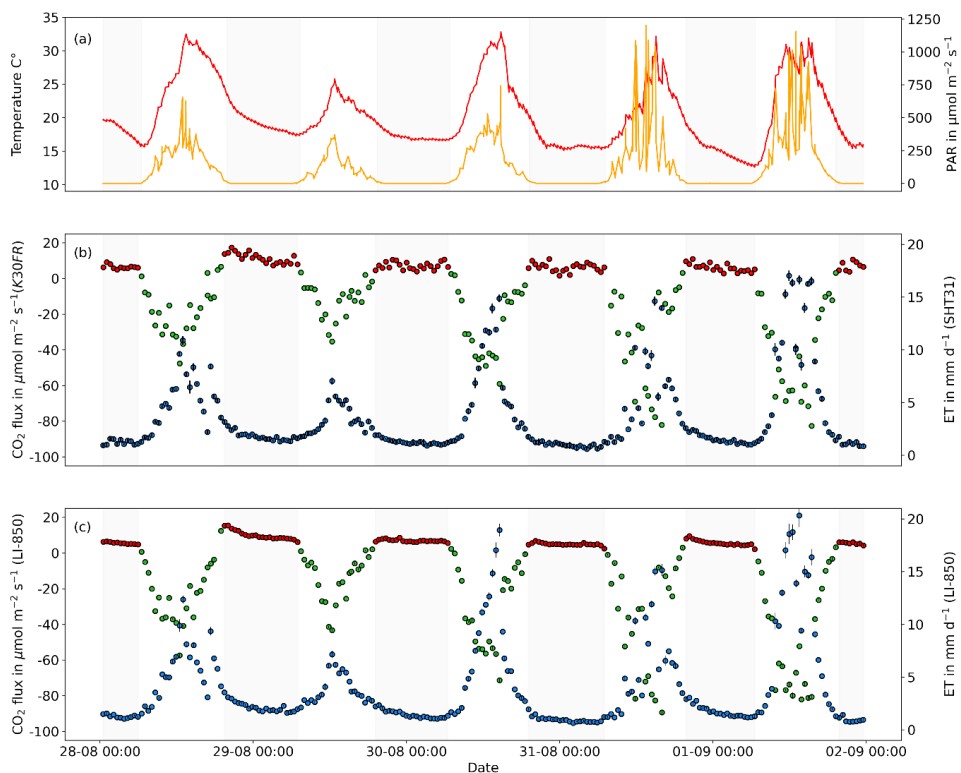


**Figure 5 shows the 5-day trial conducted for the dependent mode. a) shows the air temperature ( red line) and PAR (orange line), (b) show the diurnal cycle of $CO_2$ ($R_{eco}$: red points, NEE: green points) and ET fluxes (blue points) measured with low cost sensors ($CO_2$: K30 FR and ET: SHT31). (c) show the diurnal cycle of $CO_2$ ($R_{eco}$: red points, NEE: green points) and ET fluxes (blue points) measured with an infrared gas analyzer (LI-820, LI-COR Inc., Lincoln,**
**USA). The gray shaded areas represent the nighttime. Error bars indicate calculated fluxes error.**




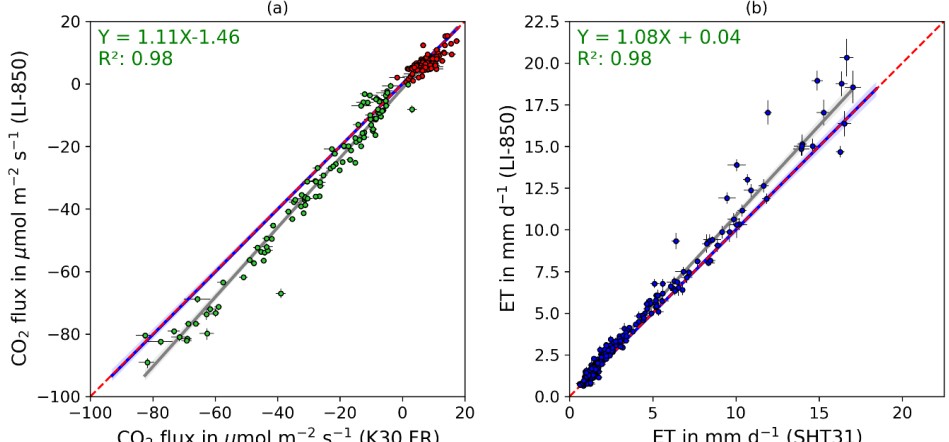

**Figure 6: 1:1 agreement between (a) $CO_2$ fluxes (Reco: red points, NEE: green points) measured with an infrared gas analyzer (LI-850, LI-COR, USA) and low cost NDIR sensor (K30 FR). As well as (b) ET fluxes (blue points) measured with an infrared gas analyzer (LI-820, LI-COR Inc., Lincoln, USA) and low cost RH sensor (SH31). The dashed red line indicates the 1:1 agreement. The grey line shows the linear regression of the measured $CO_2$ and ET fluxes, while**

**the grey shaded area represents the respective confidence band of the regression line. The blue line shows the linear regression of the corrected measured $CO_2$ and ET fluxes, while the blue shaded area represents the respective confidence band of the regression line. Error bars indicate calculated flux error.**

### 3.2.2. Dependent mode

The performed validation experiment, testing multiple greenhouse coffins with different treatments in dependent

mode, proved that by connecting multiple greenhouse coffins via a low cost multiplexer to a single infrared gas analyzer, $CO_2$ and ET fluxes can be fully automatized measured in a reliable and accurate manner. During the non-stop, five days validation experiment for the dependent mode, the tested greenhouse coffins and the used low cost Multiplexer no system errors occurred, and they worked reliably. Thus, out of 237 conducted automatic measurements, more than 75 % and 99% passed the flux calculation algorithm for $CO_2$ and ET for the treatments

with no plants (empty chamber), respectively. At the same time, 99% passed the flux calculation algorithm for $CO_2$ and ET for the other two treatments involving Sorghum and Maize, respectively. The limited number of valid $CO_2$ fluxes for the treatments with no plants can be attributed to the absence of significant changes in $CO_2$ fluxes during the measurement period. Consequently, many fluxes did not meet the IQR criteria set by the R module used for analysis. Moreover, The $CO_2$ fluxes showed no significant difference to zero (pairwise Wilcoxon signed-

rank test, $p > 0.05$), which indicates the absence of cross-contamination due to the Multiplexer (Fig.7). The $CO_2$ and ET fluxes from the greenhouse coffins containing Sorghum and Maize exhibited distinct diurnal patterns. Both treatments showed low fluxes during nighttime and higher ET fluxes and $CO_2$ uptake during daytime, clearly followed monitored environmental parameters with higher ET fluxes and $CO_2$ uptake with higher PAR and a higher $R_{eco}$ (nighttime measurements) with higher air temperatures (Fig. 7). Notably, Sorghum treatment showed

higher ET fluxes and $CO_2$ uptake compared to Maize treatment. This disparity can be explained by variations in transpiration as well as physiological responses to environmental conditions between these two plants (Farré and Faci 2006). The results highlight the system's ability to detect the diurnal cycles of $CO_2$ and ET for different treatments. This feature is highly advantageous for greenhouse studies as it allows researchers to focus on specific conditions or treatments while keeping the complexity of uncontrolled conditions, such as mesocosm experiments.

(Zaman et al. 2021) and (Bréchet et al. 2021) demonstrated the benefits of high-frequency measurements for





monitoring gas fluxes from different treatments; however, their studies were conducted under field conditions using commercial multiplexers. Furthermore, the system's capacity to link multiple greenhouse coffins to one gas analyzer and carry out measurements automatically serves to cut down on the cost as well as time that would otherwise be spent in such experiments. Finally, the choice between stand alone, fully low cost based mode and

Multiplexer connected cost efficient connected system allows for large degree of flexibility when planning experiments in terms of the target fluxes to analyzed (e.g. only $CO_2$ and $H_2O$ compared to other trace gases or stable isotope analysis, where low cost sensors are not available).

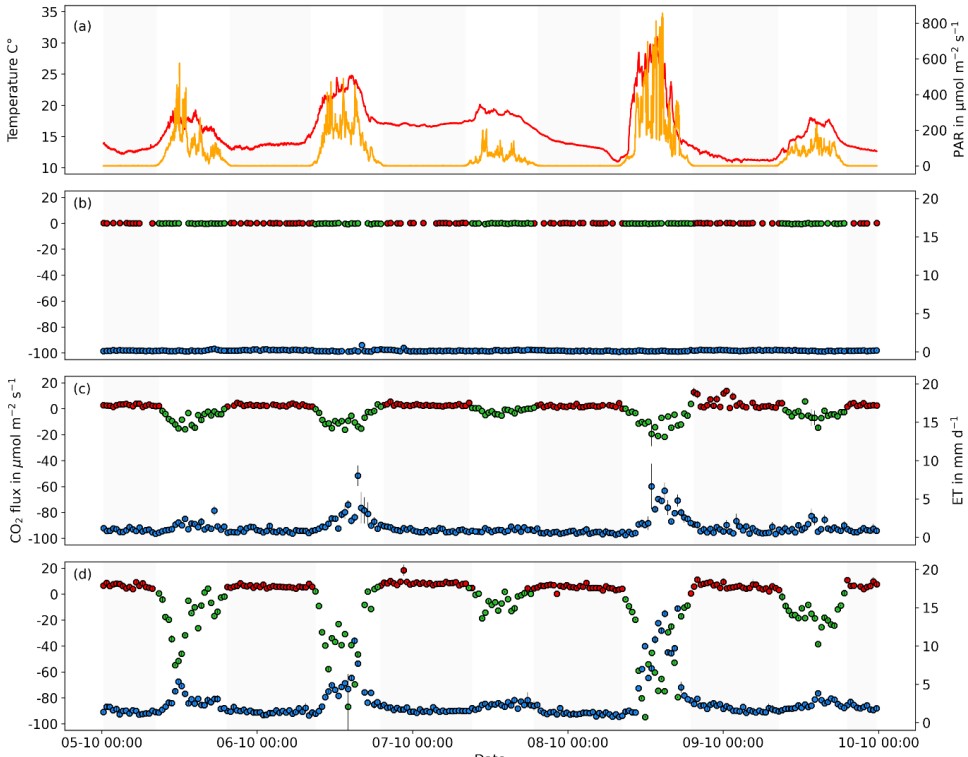

**Figure 7 shows the 5-day trial conducted for the dependent mode. a) shows the air temperature (red line) and PAR (orange line), (b, c, and d) show the diurnal cycle of $CO_2$ ( $R_{eco}$: red points, NEE: green points) and ET fluxes (blue**
**points) measured with an infrared gas analyzer (LI-850, LI-COR, USA) for three different chambers (a: without plant, b: Maiz plant, d: sorghum plant). The gray shaded areas represent the nighttime. Error bars indicate calculated fluxes error.**

## 4.    Conclusions and implications for further use:

The presented novel, low cost, automatic soil-plant enclosure system allows for an accurate and precise continuous
monitoring of gaseous exchange fluxes during pot or mesocosm experiments. This was exemplarily shown during a greenhouse pot experiment for $CO_2$ and ET fluxes of maize and sorghum. Performed system validation proved that, after calibration, $CO_2$ and ET fluxes can be determined accurately and precisely using low cost NDIR and RH sensors (independent mode). However, more importantly, by adding a low cost multiplexer to the enclosure system, other GHGs can be measured as well through adding a gas analyzer and measuring the "greenhouse coffins" in row (dependent mode). Both modes allow for cost-effective, high-temporal-resolution measurements



of soil-plant gas exchange across various treatments. In addition, the low cost modular character of the system allows for multiple further enhancements such as:

I. Parallel, high-resolution measurements of various gases such as $CO_2$, $CH_4$, $N_2O$, and $H_2O$ or also stable isotopes through combining high and low cost sensors thus allowing to determine water use efficiency, net system carbon exchange as well as full GHG balances.

II. Integrating proximal sensing of crop health and development using available low cost measurement systems to detect spectral crop indices such as NDVI or RVI.

In summary, the developed and presented system can be a valuable tool for conducting greenhouse experiments, particularly those with a high level of complexity (e.g., mesocosm experiment), allowing for holistically testing the dynamic responses of plants to various treatments and conditions while significantly reducing the required cost and labor

## 5. Code and data availability:
The data and code referred to in this study are publicly accessible at DOI:

## 6. Author contributions:
MH ,WA and MD conceptualized and developed the system and codes. WA carried out the sealing and validation experiments. WA, MH, MD and JS wrote and prepared the manuscript with contributions from all co-authors. All authors have reviewed and agreed to the final version of the paper.

## 7. Competing interests:
The contact author has declared that none of the authors has any competing interests.

## 8. Acknowledgments
Special thanks go to Andrea Hoppe for her help in the greenhouse during the system's construction.

## 9. Financial support
This research has been supported by the Leibniz Society Germany for their funding through the Leibniz Cooperative Excellence program (K378/2021) awarded to Joerg Schaller. The Open Access Fund of the Leibniz Association funded the publication of this article.

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
