# Peer review of "Technical note: A low cost, automatic soil-plant-atmosphere enclosure system to investigate CO2 and ET flux dynamics."

_EGUsphere, 2024_

## Referee Comment (RC1)

**General comments**

- The abstract is simply (e.g. accessibly, well) written, which for a gas exchange paper is a real breath of fresh air! Sometimes they are obtusely complicated which makes them hard to access, well done
- I think I have a few points on the 'broader implications' statements at the start; I'd add some specificity or remove them altogether. They feel somewhat unnecessarily broad.
- In general, I'd suggest working to change the passive voice (e.g. line 150-153) to active voice throughout (e.g. line 150 = "We measured the change in CO2 concentration after injection by connecting an infrared CO2 gas analyzer to the inlet and outlet of the sealed Greenhouse Coffin." Another example, line 159: "We conducted a greenhouse experiment to test the accuracy and precision of the low-cost sensors, as well as the overall capability of the greenhouse coffins system in independent mode." Etc.
- I made some specific notes below, but I think the paper would be stronger if it had a more focused introductory narrative. The current organization sets up an unnecessary contrast between the greenhouse coffins and existing DIY chamber systems, rather than showing that this system builds upon recent developments in DIY chamber systems as a complement (not a direct comparison, especially given the difference in application and the same sensor used in many DIY chambers these days). I have given suggestions on reorganized narrative that the authors might consider.
- I think that a last pass for sentence fragments and overly-long sentences that could be made more efficient and readable by splitting in half is in order, as the authors write very well but in some spots in a verbose way!
- This is a really cool study that fits beautifully into the growing body of literature on DIY gas sensing devices, and I love that the authors show how it will work in the greenhouse space specifically for manipulative experiments that can be applied to real-world scenarios. The authors should pump up that part of their narrative as it is quite cool!

**List of technical corrections, specific comments by location:**

- Would suggest making the first sentence more efficient *and* more germane to the actual paper's take-home by combining with the second: "Agricultural systems are particularly vulnerable to the more frequent, less predictable extreme weather events (e.g. droughts, heat waves) wrought by climate change (refs)." This kind of phrasing elimates the super-wide "funnel" at the start of the paper which is perhaps too wide for this paper's scope; yes it's true that climate change is threatening ecosystem function, but for the purposes of this study, we all already know that are want to know why ag systems in particular are the focus. (section 40)
- Section 45: I think the authors would behoove themselves to reorganize a little here. I think the 'threat' in the paper is climate change, though what it really should be is 'agricultural systems being both a source and a sink for greenhouse gases in a climate-changed world'. I suggest the authors do some (very slight, truly!) massaging of the narrative arc in this first paragraph to refocus (see above, for example). E.g., proposed rearranged 'flow' of narrative in this paragraph:
  - Agricultural systems are threatened by the changing weather patterns associated with rampant climate change
  - What is more, ag systems have the potential to both contribute to (refs) and mitigate (refs) greenhouse gas emissions depending on the practices in place and the environmental contexts of the systems.

- To best mitigate the harms of extreme weather (esp. drought, heat waves) and to characterize the potential for agricultural fields to decrease or even reverse GHG emissions, it is essential to better monitor (and thus understand) gas and water fluxes between those systems and the atmosphere.
- 52-55: "However, manual chambers require intensive labor to use at large scales and resolutions. In addition, commercial gas analyzers (not to mention the multiplexors and auto- or semi-automatic chambers associated with automatic systems) themselves are extremely expensive, presenting significant barriers to extensive chamber-based flux research, particularly in the relatively understudied global South."
- I think this needs rephrasing in light of the statements above.  Perhaps:
  - "Mesocosm-scale experiments, performed in greenhouses or climate chambers, allow researchers to mimic the *in situ* environmental conditions of many different settings, and provide the opportunity to variably manipulate those conditions within a single study site. In this way, researchers can explore the impacts of precisely isolated environmental treatments, bridging the gap between lab-based studies of single plants and field-based studies and facilitating a more nuanced understanding of ecological dynamics."
  - I will also say that I think if this is the driving thrust of the argument, the introduction should be re-framed. Right now there is a lot of content on the difficulties of field-based gas flux work given the scope/scale of those studies, resulting in a lack of study on global South conditions. But then, we move to the utility of greenhouse/mesocosm experiments, which can bridge the gap between field and lab. Which is it?  I think that the current setup should be adjusted to follow the structure I suggest above for P1, and be followed by, in P2:
    - However, it is challenging to study the effects of climate change on agricultural GHG dynamics given the difficulties inherent to field-based (high variability, environmental noise, the labor and cost associated with large-scale, high-resolution data collection and equipment) and lab-based (lack of environmental context, lack of replicability, the high cost of equipment) research on plant-soil systems.
    - Mesocosm-scale experiments located in greenhouses or climate controlled chambers therefore provide a middle ground, bridging the gap between lab and field studies by allowing for high replication, tightly controlled and isolated environmental treatments, and the ability to monitor plants within a context similar to that of their *in situ* environment.
  - Then, the next paragraph (P3) can go into the recent advances in DIY devices for GHG exchange research (without needing to discuss gap filling, which creates an artificial divide between your innovation and the current existing ones, esp. given that most of the those could easily be adapted to mesocosm experiments so it's not useful to suggest they can't. Your innovation measures something specific, the net GHG flux of a whole patch of soil/plants!  This is different and thus not directly comparable as currently suggested in line 72.
    - E.g., "In recent years, researchers have been increasingly developing low-cost devices for chamber-based gas-exchange systems using a do-it-yourself (DIY) approach. These DIY systems reduce the generally high cost per device, allowing for higher replicability than has been previously possible using commercial systems. They leverage...such as the "Fluxbots".

> To expand the application space of such DIY devices to the mesocosm scale, we have developed and validated the "Greenhouse Coffin", a novel..."

- 80: highlighted words that can be deleted in green, here and throughout
- 80: spell out "relative humidity (RH)" here and use RH for the remainder
- 80: not sure what 'based' means here in the context, apologies! Highlighted to flag it for the authors to confirm
- 82-83: suggest rephrasing to, "Furthermore, we tested a DIY, low-cost multiplexer's ability to link multiple greenhouse coffins to one commercial gas analyzer." ← since you're testing the multiplexor, not the system per se!
- 92: what does "Arduino Uno-like" mean? Isn't the ATmega a kind of mcu that can be associated with an Arduino board? I would clarify what you mean here otherwise I think it'll cause confusion.
- 107: "thus enabling *researchers* to chain each greenhouse coffin *together* to a single gas analyzer"
- 115: see note above on line 92 re: microcontroller specs; this is a little bit confusing!
- 166: in what way does the Bluetooth allow for easy data access? I'd love a few more details on how this works aka what format is the data in, how does it get transmitted over Bluetooth, etc.! it seems cool ☺
- Fig. 1: I think the labels on the two modes are incorrect; I think the left needs to be *independent* mode and the right needs to be *dependent*, right? the legend is correct if so, just the labels are off!
- 142: ha! This is awesome ☺
- 180 section: I suggest a table with the gas constants listed for easy access for readers looking to replicate your data processing method!
- 242: this wording is a little awkward and fumbly; I also think it's probable that you'll want to say "demonstrated" over "proved". Maybe, "The validation experiment, performed continuously over five days using a single greenhouse coffin in independent mode, demonstrated that CO2 and ET fluxes can be measured reliably and accurately in a fully automated chamber using low-cost sensors."
  - Remove highlighted sentence in 243-244
  - "...using low-cost sensors. Out of 223 automated measurements..."

---

## Author Comment (AC1)

We thank the editor and the anonymous reviewer 1 for their valuable comments which will substantially improve our revised manuscript entitled: "Technical note: A low cost, automatic soil-plant-atmosphere enclosure system to investigate $CO_2$ and ET flux dynamics.". We have carefully addressed all comments of both reviewers. Please note the color code in our point-by-point answer below: (I.) reviewer comments are presented in black; (II.) our replies are in green; (III.) manuscript passages including suggested changes are presented in italic and gray

**General comments:**

The abstract is simply (e.g. accessibly, well) written, which for a gas exchange paper is a real breath of fresh air! Sometimes they are obtusely complicated which makes them hard to access, well done.

I think I have a few points on the 'broader implications' statements at the start; I'd add some specificity or remove them altogether. They feel somewhat unnecessarily broad.

1.) In general, I'd suggest working to change the passive voice (e.g. line 150-153) to active voice throughout (e.g. line 150 = "We measured the change in CO2 concentration after injection by connecting an infrared CO2 gas analyzer to the inlet and outlet of the sealed Greenhouse Coffins." Another example, line 159: "We conducted a greenhouse experiment to test the accuracy and precision of the low-cost sensors, as well as the overall capability of the greenhouse Coffins system in independent mode." Etc.

We will change the passive to active voice as suggested throughout the entire MS.

2.) I made some specific notes below, but I think the paper would be stronger if it had a more focused introductory narrative. The current

organization sets up an unnecessary contrast between the greenhouse Coffins and existing DIY chamber systems, rather than showing that this system builds upon recent developments in DIY chamber systems as a complement (not a direct comparison, especially given the difference in application and the same sensor used in many DIY chambers these days). I have given suggestions on reorganized narrative that the authors might consider.

We agree with your suggestions and the revised manuscript will introduce our method as complementary to existing developments and presenting recent developments in a balanced way. Please see the joined reply to comment 9.

3.) I think that a last pass for sentence fragments and overly-long sentences that could be made more efficient and readable by splitting in half is in order, as the authors write very well but in some spots in a verbose way!

We will thoroughly rework the entire MS with focus on readability by (i) shorted/split sentences where possible, (ii) using active voice and (iii) eliminating redundant/unnecessary wording.

4.) This is a really cool study that fits beautifully into the growing body of literature on DIY gas sensing devices, and I love that the authors show how it will work in the greenhouse space specifically for manipulative experiments that can be applied to real-world scenarios. The authors should pump up that part of their narrative as it is quite cool!

Thank you for the positive feedback which we greatly appreciate. We will try to boost this narrative.

**List of technical corrections, specific comments by location:**

5.)    Would suggest making the first sentence more efficient and more germane to the actual paper's take-home by combining with the second: "Agricultural systems are particularly vulnerable to the more frequent, less predictable extreme weather events (e.g. droughts, heat waves) wrought by climate change (refs)." This kind of phrasing eliminates the superwide "funnel" at the start of the paper which is perhaps too wide for this paper's scope; yes, it's true that climate change is threatening ecosystem function, but for the purposes of this study, we all already know that are want to know why ag systems in particular are the focus. (section 40).
Done as suggested. Please see the joined reply to comment 4.

6.)    **Section 45**: I think the authors would behoove themselves to reorganize a little here. I think the 'threat' in the paper is climate change, though what it really should be is 'agricultural systems being both a source and a sink for greenhouse gases in a climate changed world'. I suggest the authors do some (very slight, truly!) massaging of the narrative arc in this first paragraph to refocus (see above, for example). E.g., proposed rearranged 'flow' of narrative in this paragraph:

    o    Agricultural systems are threatened by the changing weather patterns associated with rampant climate change.

    o    What is more, ag systems have the potential to both contribute to (refs) and mitigate (refs) greenhouse gas emissions depending on the practices in place and the environmental contexts of the systems.

    o    To best mitigate the harms of extreme weather (esp. drought, heat waves) and to characterize the potential for agricultural fields to

decrease or even reverse GHG emissions, it is essential to better monitor (and thus understand) gas and water fluxes between those systems and the atmosphere.

Done as suggested and changed in the MS as follows: *"Agricultural systems are particularly vulnerable to the more frequent, less predictable extreme weather events (e.g. droughts, heat waves) wrought by climate change (Altieri et al. 2015; Ummenhofer and Meehl 2017). Moreover, agricultural systems have the potential to both contribute to (Tubiello et al. 2013; Chataut et al. 2023) and mitigate (Lal 2004; Powlson et al. 2016) greenhouse gas (GHG) emissions, influenced by the practices implemented and the specific environmental contexts in which they operate. Therefore, to best mitigate the harms of extreme weather (especially drought and heat waves) and to characterize the potential for agricultural fields to decrease or even reverse GHG emissions, it is essential to better monitor (and thus understand) gas and water fluxes between those systems and the atmosphere (Zhang et al. 2002; Joshua B. Fisher et al. 2017)."*

7.)    **52-55**: "However, manual chambers require intensive labor to use at large scales and resolutions. In addition, commercial gas analyzers (not to mention the multiplexors and auto- or semi-automatic chambers associated with automatic systems) themselves are extremely expensive, presenting significant barriers to extensive chamber-based flux research, particularly in the relatively understudied global South." I think this needs rephrasing in light of the statements above. Perhaps:

o    "Mesocosm-scale experiments, performed in greenhouses or climate chambers, allow researchers to mimic the in situ environmental conditions of many different settings, and provide the opportunity to variably

manipulate those conditions within a single study site. In this way, researchers can explore the impacts of precisely isolated environmental treatments, bridging the gap between lab-based studies of single plants and field-based studies and facilitating a more nuanced understanding of ecological dynamics.".

Done as suggested. Please see the joined reply to comment 8.

8.)     I will also say that I think if this is the driving thrust of the argument, the introduction should be re-framed. Right now there is a lot of content on the difficulties of field-based gas flux work given the scope/scale of those studies, resulting in a lack of study on global South conditions. But then, we move to the utility of greenhouse/mesocosm experiments, which can bridge the gap between field and lab. Which is it? I think that the current setup should be adjusted to follow the structure I suggest above for P1, and be followed by, in P2:

- However, it is challenging to study the effects of climate change on agricultural GHG dynamics given the difficulties inherent to field-based (high variability, environmental noise, the labor and cost associated with large-scale, high-resolution data collection and equipment) and lab-based (lack of environmental context, lack of replicability, the high cost of equipment) research on plant-soil systems.
- Mesocosm-scale experiments located in greenhouses or climate controlled chambers therefore provide a middle ground, bridging the gap between lab and field studies by allowing for high replication, tightly controlled and isolated environmental

treatments, and the ability to monitor plants within a context similar to that of their in situ environment.

Done as suggested and changed in the MS as follows:

*"Chamber-based systems (automatic or manual) in conjunction with high temporal resolution gas analyzers are one of the most common techniques for directly measuring $CO_2$ and evapotranspiration (ET), providing precise data on a leaf to plot scale and allowing to assess small scale heterogeneity (Smith et al. 2010; Dubbert et al. 2014; Riederer et al. 2014). However, it is challenging to study the effects of climate change on agricultural GHG dynamics given the difficulties inherent to both field-based and laboratory based research on soil-plant-atmosphere systems. Field based research comes at the expanse of high variability, environmental noise and the labor and cost associated with large-scale, high-resolution data collection and equipment, whereas lab-based is limited by a lack of environmental context and replicability beside the high cost of equipment (Savage and Davidson 2003, Sun, X. et al. 2013; Martin et al. 2017; Blackstock et al. 2019). Mesocosm-scale experiments on the other hand, performed in greenhouses or climate controlled chambers, allow researchers to mimic the in situ environmental conditions of many different settings, and provide the opportunity to variably manipulate those conditions within a single study site. In this way, researchers can explore the impacts of precisely isolated environmental treatments, bridging the gap between lab-based studies of single plants and field-based studies and facilitating a more nuanced understanding of ecological dynamics. (Riebesell et al. 2013; Stewart et al. 2013)."*

9.)     Then, the next paragraph (P3) can go into the recent advances in DIY devices for GHG exchange research (without needing to discuss gap filling, which creates an artificial divide between your innovation and the current existing ones, esp. given that most of the those could easily be adapted to mesocosm experiments so it's not useful to suggest they can't. Your innovation measures something specific, the net GHG flux of a whole patch of soil/plants! This is different and thus not directly comparable as currently suggested in line 72.

- E.g., "In recent years, researchers have been increasingly developing low cost devices for chamber-based gas-exchange systems using a do-it yourself (DIY) approach. These DIY systems reduce the generally high cost per device, allowing for higher replicability than has been previously possible using commercial systems. They leverage…such as the "Fluxbots". To expand the application space of such DIY devices to the mesocosm scale, we have developed and validated the "Greenhouse Coffin", a novel…"

Done as suggested and changed in the MS as follows:

*"In recent years, researchers have been increasingly developing low cost devices for chamber-based gas-exchange systems using a do-it yourself (DIY) approach. These DIY systems reduce the generally high cost per device, allowing for higher replicability than has been previously possible using commercial systems (Fisher and Gould 2012; D'Ausilio 2012). They leverage affordable microcontrollers and sensors to build custom measurement tools designed for specific research needs. By integrating sensors for $CO_2$ and/or ET with microcontrollers, researchers were able to develop portable, precise, and cost-effective devices for monitoring $CO_2$ and ET fluxes, such as Macagga et al. (2024) and Bonilla-Cordova et al.*

*(2024). Others went a step further and developed fully automated measurement systems to determine $CO_2$ efflux, such as the "Fluxbots" (Forbes et al. 2023).*

*To expand the application space of such DIY devices to the mesocosm scale, we have developed and validated the "Greenhouse Coffins", a novel low cost automatic soil-plant enclosure system, designed to monitor $CO_2$ and ET fluxes within greenhouse experiments in a fully automatic manner. "*

10.) **80**: highlighted words that can be deleted in green, here and throughout.

We did not receive a PDF copy that has been marked by you. However, the revised version will be carefully checked to avoid unnecessary sentence fragments and words.

11.) **80**: spell out "relative humidity (RH)" here and use RH for the remainder.

Done.

12.) **80**: not sure what 'based' means here in the context, apologies! Highlighted to flag it for the authors to confirm.

We deleted the incorrect wording "their based" from the sentence, which now reads as follows:

*"Additionally, we evaluated the accuracy and precision of used low-cost NDIR $CO_2$ (K30 FR) and RH sensors (SHT31) by comparing their calculated $CO_2$ and ET fluxes with results obtained with a commercial infrared gas analyzer (LI-850, LI-COR Inc., Lincoln, USA)."*

13.) **82-83**: suggest rephrasing to, "Furthermore, we tested a DIY, low-cost multiplexer's ability to link multiple greenhouse Coffins to one commercial gas analyzer." since you're testing the multiplexor, not the system per se!

Done accordingly.

14.) **92**: What does "Arduino Uno-like" mean? Isn't the ATmega a kind of mcu that can be associated with an Arduino board? I would clarify what you mean here otherwise I think it'll cause confusion.

To avoid confusion, we changed "Arduino Uno-like" for "ATmega328 Microcontroller" throughout the entire MS (Arduino Uno-like refers to a cheap clone with similar properties, which is, however, not produced by the company Arduino). In addition, we added the description column to Tab.1, which now describes all components in more detail.

15.) **107**: "thus enabling researchers to chain each greenhouse coffin together to a single gas analyzer".

Changed accordingly.

16.) **115**: see note above on line 92 re: microcontroller specs; this is a little bit confusing!

To avoid confusion, we changed "Arduino mega-like" for "ATmega2560 Microcontroller" throughout the entire MS (Arduino Uno-like refers to a cheap clone with similar properties), which is however not produced by the company Arduino). In addition, we added the description column to Tab.1, which now describes all components in more detail.

**17.) 166**: in what way does Bluetooth allow for easy data access? I'd love a few more details on how this works aka what format is the data in, how does it gets transmitted over Bluetooth, etc.! It seems cool.

Bluetooth facilitates easy data access by wirelessly transmitting data to another Bluetooth device in a text format, which can be easily read and processed by various software applications. This setup enables direct monitoring near the greenhouse coffin via a smartphone or tablet using a Serial application (e.g., Serial Bluetooth). Additionally, the microcontroller can be connected to a computer or Raspberry Pi keyboard, where the data is recorded as text, plotted, and can be monitored remotely using software like AnyDesk. We will include these details in the manuscript to provide a clearer explanation of how Bluetooth facilitates data access.

1.    **Fig. 1**: I think the labels on the two modes are incorrect; I think the left needs to be independent mode and the right needs to be dependent, right? The legend is correct if so, just the labels are off.
We corrected it accordingly.

2.    **142**: ha! This is awesome.
…and fun.

3.    **180 section** : I suggest a table with the gas constants listed for easy access for readers looking to replicate your data processing method!
Please note the ideal gas constant is given in the MS as: "8.314 m³ Pa K⁻¹ mol⁻¹".

4.    **242**: this wording is a little awkward and fumbly; I also think it's probable that you'll want to say "demonstrated" over "proved". Maybe,

"The validation experiment, performed continuously over five days using a single greenhouse coffin in independent mode, demonstrated that CO2 and ET fluxes can be measured reliably and accurately in a fully automated chamber using low-cost sensors.".

o        Remove highlighted sentence in 243-244.

o        "…using low-cost sensors. Out of 223 automated measurements…".

Done as suggested.

"The validation experiment, performed continuously over five days using a single greenhouse coffin in independent mode, demonstrated that $CO_2$ and ET fluxes can be measured reliably and accurately in a fully automated chamber using low-cost sensors. Thus, out of 223 conducted automatic measurements, more than 99% passed the flux calculation algorithm for $CO_2$ and ET, respectively. "

---

## Author Comment (AC2)

We thank the editor and the anonymous reviewer 2 for their valuable and constructive comments which substantially improve our revised manuscript entitled: "Technical note: A low cost, automatic soil-plant-atmosphere enclosure system to investigate $CO_2$ and ET flux dynamics.". We have carefully addressed all comments of both reviewers. Please note the color code in our point-by-point answer below:

(I.) reviewer comments are presented in black; (II.) given answers are presented in green; (III.) manuscript passages including suggested changes are presented in italic and gray

The authors present a description of a low-cost mesocosm CO2 flux and ET measurement system. The basic idea of the manuscript and the measurement system is good, as there is a large need for low-cost instrumentation for scientific studies in the developing world; the authors well point out this reasoning for their study. In its current form the manuscript is, however, not publishable without major revisions and further tests.

1.)     The level of technical detail within the manuscript is a bit too varying; on the one hand, the Mosfet (the meaning of which many researchers probably are not familiar with!) is described down to a component code and the precise ohm numbers of the resistors, but the manufacturer and model of the linear actuator or the data logging shield, of which there are many available, are not disclosed; neither are the properties of the air-mixing and ventilation fans disclosed: what volume of air do they move per minute.

We understand the importance of providing consistent technical detail throughout the manuscript. In response, we will provide more detailed technical specifications and properties of the components in Table 1, including the manufacturer and model information for the linear actuator, data logging shield, and the air-mixing and ventilation fans, along with their respective specifications, such as Volumetric flow. In addition, we carefully checked the manuscript to avoid presenting in varying levels of details (see also replies to specific comments below).

The schematics in Fig. 2 are of little use: at first sight, they appear detailed, but the small scale of the images makes deciphering the precise connections difficult or impossible. A proper schematic drawing (describing which pins on the microcontroller are connected to which pins on the relay board, for example) should be made available along the Arduino microcode to enable readers to build systems of their own.

We agree and will provide more detailed schematics to make it easier to follow. Additionally, pin connections are provided now within the schematic (before only given within the Arduino code, which is available through a DOI link). We will update the manuscript accordingly to include these details, ensuring that readers have all the necessary information to build and understand the system. Please see the updated figure below.

[Figure]

**Figure 2** Schematic representation of the wiring of one Greenhouse Coffin in the dependent mode.

2.) The design of the "coffin" is not well described. It's not clear whether it's a ready-made design by the Polish firm Romid (if, then what order code?), or constructed by the authors; and if it's self-constructed, how the door and sliding window are constructed (hinges, rails, etc?), how is a tight seal ensured when the window is closed, etc. These inconsistencies make it unclear whether the manuscript is meant to be a general description of the principles of a measurement system or a blue print. The authors should decide which approach they want to use.

We apologize for the confusion. The design of the Greenhouse coffin was developed by us and all electronic parts were assembled by us. The PVC construction of the coffin, including most drillings, was done by Romid, who received a detailed construction schematic from us for the customized construction. To make that clear, a description column is added to Table 1. which states now the following regarding the chamber body:

*"Design by authors and customized build of the PVC construction (180\*40\*60 cm) via Romid company.".*

Hence, no order code exists. However, the dimensions of the greenhouse coffin body might differ for other purposes (smaller body for smaller plants, bigger body for bigger plants). Thus, for the working principle of the presented coffin system, the given dimension is an example of how it could be done rather than a fixed standard of how it must be done as long as each chamber design is tested for proper sealing and ventilation.

Tight guiding rails were used as a mechanism for proper window closure and sealing, which we added to the MS as follows:

*" The front door is equipped with a sliding window mechanism, which is opened and closed by a linear actuator moving it along guiding rails."*

3.) A smaller issue, on line 179, I think the Li-850 already corrects its readings for $H_2O$ interference? The authors should double-check this (and present the formula for $H_2O$ correction if they need to apply one!).

LI-850 does indeed correct its readings for Instrument Cross-sensitivity. However, we referred to Dilution by Foreign Gases (Hupp, J. et al. 2011). To avoid any misunderstanding based on our wording, we rephrased the sentence. in addition, the used correction function is given as follow:

*"Additionally, the $CO_2$ concentrations measured with the LI-850 were corrected for the changes in water vapor during each chamber measurement (correction for dulation by foreign gas; Webb et al. 1980;Hupp, J. et al. 2011) Eq.(1):*

$$C_g^{wr} = C_g^{ws} \frac{1 - w_r/1000}{1 - w_S/1000} \tag{1}$$

*Where $C_g^{wr}$ is the mole fraction of $CO_2$ in the sample (µmol/mol) corrected to the water vapor content of the reference measurement $w_r$ (mmol/mol), $C_g^{ws}$ is the mole fraction of $CO_2$ measured in the sample (µmol/mol), and $w_S$ is the water vapor content in the sample (mmol/mol). "*

4.)    I find the flux calculation method somewhat strange. Using a variable-size moving window and discriminating against larger temperature changes would seem to prioritize moments when the sun is occluded (low temperature rise) or in the case of constant sunlight cases when the temperature difference between inside and outside is already high (higher outflux of heat lessens the T rise within the chamber), or short fitting times. Instinctively I'd prefer a more constant approach to the fitting, e.g. decide that the fitting time is 4 minutes, leaving 1 minute out from the start. This is not a critical issue here, but if the authors plan to use the system for some actual measurement campaign, they should further examine how gas fluxes are estimated in closed-loop setups in other studies.

A discrimination against larger temperature changes is rather unlikely since < 5% of all fluxes showed an increase more than the used 1.5°C threshold and none exceeded 2°C. Various studies have employed different methods to limit the influence of temperature increases in closed-loop systems on calculated fluxes, such as (i) implementing cooling systems (Beetz, S. et al, 2013), (ii) shortening chamber closure time (Grace, P. R. et al., 2020), or focusing on specific parts of the measurement where temperature changes do not exceed a certain threshold (Leiber-Sauheitl, K. et al. 2014;

Hoffmann, M. et al. 2015). A variable moving window has the advantage of automatically finding the optimum time window for fitting a linear regression to nighttime Reco (usually longer) as well as daytime NEE measurements (usually shorter). A fixed measurement window of e.g. 4 minutes could potentially discriminate against non clear sky conditions (characterized by a fast change between full sunlight and cloud-induced shading), with an immanent effect on $CO_2$ gas exchange. For example, Koskinen, M. et al (2014) found that determining the best interval based only on $CO_2$ concentration is not possible; the stability of the flux should also be considered.

5.) The method for testing the sealing of the system is seriously lacking: a smoke bomb creates aerosols, which are multiple orders of magnitude larger than the CO2 and H2O molecules which are the object of measurement here. The proper way of estimating leak (which is inavoidable in a system like this!) is to create a large mixing ratio of CO2, such as 1000 or 2000 ppm, in the chamber and then monitor the development of the mixing ratio within the chamber compared to the surrounding mixing ratio (ppm s-1 delta_ppm-1, where ppm is the mixing ratio within the chamber and delta_ppm is the difference between the outside and inside). Thus, a solid estimate of the proportion of air exchanged between the measurement system and the ambient atmosphere can be made. The same method can be used to estimate the rate of leakage between the chambers in multi-chamber mode, without the need to use a semi-random factor such as plants in the process. The method of leak evaluation chosen by the authors is not proper for the job. A small difference between mixing ratios on the inside and outside makes leaks nearly undetectable.

While a smoke bomb test does not prove air tightness, it nicely indicates serious air leakage (and more importantly, where exactly it occurs) from the Greenhouse coffin. To better emphasize this and avoid misunderstandings, we will change it in the MS as following:

*"To assess for a serious leakage from the Greenhouse Coffin (and more importantly where exactly on the construction it occurs), a smoke bomb was used as suggested by (Hoffmann et al. 2018) and which was also used by*

*(Olfs et al. 2018) for the leakage test on their developed chamber design to measure nitrous oxide.''*

In addition, we extended the magnitude of our leakage test with $CO_2$ injections, as requested, to a larger mixing ratio of $CO_2$ (1000 ppm) and will use it to update Figure 4 accordingly. The test results showed no significant difference (pairwise Wilcoxon signed-rank, $p> 0.05$) between $\Delta CO_2$ measured by the LI-850 and the calculated mixing ratio. As previously concluded, this suggests the proper sealing of the coffin. See updated Figure 4 below:

[Figure]

**Figure 4**:1:1 agreement between the mixing ratio and the measured $\Delta CO_2$ concentration change expressed as in ppm, was obtained during the laboratory validation.

6.)    Another thing missing is an estimation of how the enclosure affects air temperature: it's a rather well-closed system without any cooling aside the ventilation, so I suspect that the temperature inside can get quite high on a sunny day.

To prevent temperature increases inside the Greenhouse Coffin, we selected ventilation fans with a volumetric flow rate of 76.4 m³/h, allowing for a complete air exchange within 20 seconds during the opening period (this information is now added to the MS). Thus, the average temperature difference between the inside and outside of the greenhouse coffin during the measurement period was $< 0.25°C$.

7.)    Currently there is an increasing interest in non-CO2 GHG:s (esp. N2O, CH4) emitted from and/or consumed by plants. These can be tricky to measure as the mixing ratios are low and this makes the estimation of leaks even more important. Another important thing is that the materials used for constructing measurement setups, such as rubbers, plastics, glues etc. can emit the GHGs themselves or volatile organic compounds that can mimic or mask these GHGs in the measurement devices. An estimation of the blank flux rate of other greenhouse gases than H2O and CO2 would be very interesting; or the authors should include mention of the need for such a test in their first enhancement proposal (ll. 343-345). We agree, for non-$CO_2$ GHGs the sealing of the Greenhouse coffin needs to be still tested in further experiments. We also agree that for BVOCs the used material needs to be thoroughly chosen and tested for potential outgassing. Hence, we will include the following in the MS:

*"Parallel, high-resolution measurements of various gasses such as $CO_2$, $CH_4$, $N_2O$, and $H_2O$ or also stable isotopes through combining high and low cost sensors thus allowing to determine water use efficiency, net system carbon exchange as well as full GHG balances. However, to ensure proper sealing, thorough sealing tests are crucial, particularly since gases like $N_2O$ and $CH_4$ have low mixing ratios. Additionally, careful consideration must be given to the materials used in the construction, as they may emit GHGs or volatile organic compounds that could affect the accuracy of measurements."*

**References:**

Soneye, O. O., Ayoola, M. A., Ajao, I. A., & Jegede, O. O. (2019). Diurnal and seasonal variations of the incoming solar radiation flux at a tropical station, Ile-Ife, Nigeria. Heliyon, 5(5).

Beetz, S., Liebersbach, H., Glatzel, S., Jurasinski, G., Buczko, U., & Höper, H. (2013). Effects of land use intensity on the full greenhouse gas balance in an Atlantic peat bog. Biogeosciences, 10(2), 1067-1082.

Leiber-Sauheitl, K., Fuß, R., Voigt, C., & Freibauer, A. (2014). High CO 2 fluxes from grassland on histic Gleysol along soil carbon and drainage gradients. Biogeosciences, 11(3), 749-761.

Grace, P. R., van der Weerden, T. J., Rowlings, D. W., Scheer, C., Brunk, C., Kiese, R., ... & Skiba, U. M. (2020). Global Research Alliance N2O chamber methodology guidelines: Considerations for automated flux measurement. Journal of Environmental Quality, 49(5), 1126-1140.

Hoffmann, M., Jurisch, N., Borraz, E. A., Hagemann, U., Drösler, M., Sommer, M., & Augustin, J. (2015). Automated modeling of ecosystem CO2 fluxes based on periodic closed chamber measurements: A standardized conceptual and practical approach. Agricultural and Forest Meteorology, 200, 30-45.

Hupp, J. (2011). The importance of water vapor measurements and corrections. LI-COR Biosciences Inc. Application Note, 129, 8.

Koskinen, M., Minkkinen, K., Ojanen, P., Kämäräinen, M., Laurila, T., & Lohila, A. (2014). Measurements of CO 2 exchange with an automated chamber system throughout the year: challenges in measuring night-time respiration on porous peat soil. Biogeosciences, 11(2), 347-363.

---

## Author Response (AR2)

Thank you once more for constructively reviewing our manuscript. Reviewer comments are in black, our reply is in green, and passages cited from the revised manuscript are in grey italics.

I suggest the authors either describe better the $CO_2$ injection leak estimation protocol, or for better and more useful results, conduct a proper leak test like I outlined in my previous comments.

The minimum to do is to specify the length of the observation period after $CO_2$ injection and how the reported difference between calculated and observed $CO_2$ concentrations was assessed: was the reported inside $CO_2$ concentration obtained as just the highest single $CO_2$ concentration observed after injection, or if it is the average concentration over some time window (and what the time window was), or what is it.

Of real use would be to make an estimate of how much air is exchanged between the inside and outside of the coffin per time unit, which can be estimated by measuring the $CO_2$ concentration outside the coffin and inside the coffin during a specified observation period, such as 10 minutes or a half hour. The central point here is that leakage is always present in a closed-loop system of this size, and the remedy for leak is to estimate the leak rather than pretend it's not there.

We now specify the length of the observation period following $CO_2$ injection (5 min in a 5 sec interval, which also matches max. chamber closure time during $CO_2$ and ET flux measurements in our setup. Thereby, we follow the common guidelines for closed chamber measurements (Pirk, N. et al. 2016; Maier et al. 2022). Further, we report how the difference between calculated and observed CO2 concentrations was assessed in a more clearly manner within the MS (L170-180):

*"To check for the suitability of the sliding window to sufficiently seal the Greenhouse Coffin airtight when closed and exchange air when open in its final setup (complete hardware implementation), we repeatedly injected distinct amounts of technical gas containing 1,000,000 ppm $CO_2$ ranging from 15 to 450 ml into its sealed headspace using a syringe. Prior, during, and after each injection, chamber headspace $CO_2$ concentrations were continuously recorded in a 5-second interval using an infrared $CO_2$ gas analyzer (LI-850, LI-COR Inc., Lincoln, USA) connected to the inlet and outlet of the coffin. In more detail, the following procedure was opted: (1) After the sliding door was closed and stable $CO_2$ concentrations were obtained (ca. 1 minute), (2) technical gas was injected into the chamber headspace, and $CO_2$ concentration development was recorded over the next 5 minutes before (3) the sliding door was opened again and $CO_2$ concentration depletion was monitored until stabilization (ca 1 minute). The average $CO_2$ concentration of the initial 1 minute (12 records; after closure and before injection) and last 4 minutes (48 records; after injection/stabilization and before opening) of $CO_2$ concentration records were then used to calculate the change in $CO_2$ concentration from before to after injection ($\Delta CO_2$ in ppm). In case of proper sealing of the coffin, the thus determined $\Delta CO_2$ should match the calculated mixing ratio."*

During the 5 minutes observation period following each injection (- 1 minute for stabilization (mixing of chamber air) of $CO_2$ concentrations directly after injection), measured $CO_2$ concentrations were stable and did not show a decline over measurement time. Only after the sliding door was opened again, $CO_2$ concentrations rapidly reached initial values, indicating proper/sufficient sealing during the measurement period.

While we agree that a determination of how much air is exchanged between the inside and outside of the coffin per time unit would be additionally interesting, to our understanding, this would require a box in a box setup (chamber around coffin). Only thus would concentrations around the coffin be stable enough to detect the potential minimal $CO_2$ concentration changes that can be expected and are indicated by our tests. Otherwise, the common fluctuations in $CO_2$ concentration present in a greenhouse setup (e.g., respiration by scientific staff (also those injecting the technical gas), diurnal cycle, etc.) would interfere with accurately assessing leakage at such low levels.

References:

Pirk, N., Mastepanov, M., Parmentier, F. J. W., Lund, M., Crill, P., & Christensen, T. R. Calculations of automatic chamber flux measurements of methane and carbon dioxide using short time series of concentrations. *Biogeosciences*, *13*(4), 903-912. (2016).

Maier, Martin; Weber, Tobias K. D.; Fiedler, Jan; Fuß, Roland; Glatzel, Stephan; Huth, Vytas et al.: Introduction of a guideline for measurements of greenhouse gas fluxes from soils using non-steady-state chambers. In Journal of Plant Nutrition and Soil Science 185 (4), pp. 447–461. DOI: 10.1002/jpln.202200199. (2022)